# Clustering Analysis Supports the Detection of Biological Processes Related to Autism Spectrum Disorder

**DOI:** 10.3390/genes11121476

**Published:** 2020-12-09

**Authors:** Leonardo Emberti Gialloreti, Roberto Enea, Valentina Di Micco, Daniele Di Giovanni, Paolo Curatolo

**Affiliations:** 1Department of Biomedicine and Prevention, University of Rome Tor Vergata, Via Montpellier 1, 00133 Rome, Italy; 2IMME Research Centre, Via Giotto 43, 81100 Caserta, Italy; roberto.enea@dynamiqtest.com; 3Child Neurology and Psychiatry Unit, Systems Medicine Department, University of Rome Tor Vergata, Via Montpellier 1, 00133 Rome, Italy; dimiccovalentina93@gmail.com (V.D.M.); curatolo@uniroma2.it (P.C.); 4Department of Industrial Engineering, University of Rome Tor Vergata, Via del Politecnico 1, 00133 Rome, Italy; daniele.di.giovanni@uniroma2.it

**Keywords:** autism spectrum disorder (ASD), cluster analysis, gene networks, patient similarity analytics, neurite morphogenesis, synapse assembly, connectivity

## Abstract

Genome sequencing has identified a large number of putative autism spectrum disorder (ASD) risk genes, revealing possible disrupted biological pathways; however, the genetic and environmental underpinnings of ASD remain mostly unanswered. The presented methodology aimed to identify genetically related clusters of ASD individuals. By using the VariCarta dataset, which contains data retrieved from 13,069 people with ASD, we compared patients pairwise to build “patient similarity matrices”. Hierarchical-agglomerative-clustering and heatmapping were performed, followed by enrichment analysis (EA). We analyzed whole-genome sequencing retrieved from 2062 individuals, and isolated 11,609 genetic variants shared by at least two people. The analysis yielded three clusters, composed, respectively, by 574 (27.8%), 507 (24.6%), and 650 (31.5%) individuals. Overall, 4187 variants (36.1%) were common to the three clusters. The EA revealed that the biological processes related to the shared genetic variants were mainly involved in neuron projection guidance and morphogenesis, cell junctions, synapse assembly, and in observational, imitative, and vocal learning. The study highlighted genetic networks, which were more frequent in a sample of people with ASD, compared to the overall population. We suggest that itemizing not only single variants, but also gene networks, might support ASD etiopathology research. Future work on larger databases will have to ascertain the reproducibility of this methodology.

## 1. Introduction

Autism spectrum disorder (ASD) is a heterogeneous group of neurodevelopmental disorders characterized by impaired social communication, repetitive behaviors, and restricted interests [1]. Genetic [2,3] and epigenetic [4] factors have been identified as leading actors in ASD pathophysiology, as twin studies have confirmed. A meta-analysis on 6413 twins, including affected twins, showed a heritability in families with an autistic patient of 64–91% [5]. Autism appears to be the final outcome of complex genetic and epigenetic architectures [3], with possible contributions of environmental factors [6]. However, many questions regarding interactions between genes and environment still remain unanswered [7].

In fact, in the light of more recent genetic research, it has been highlighted that thousands of genetic variants may contribute to this disorder [8], further confirming the complex heterogeneous etiology of ASD, with consequent variegated configurations of biological and behavioral characteristics [9]. This heterogeneity calls for the need to ground autism research more and more on studies and techniques which can cope with large sample sizes [10].

During the last decades, developments in gene-hunting techniques have identified several ASD associated genes, including genes that code for proteins involved in synaptic functions [8,11]. Such new technologies, including exome-wide and genome-wide interrogation, are considered effective methodologies for detecting links between a common variant located in a specific DNA region and the risk of developing ASD [2]. More recently, the largest exome sequencing study in ASD, which analyzed a cohort of 35,584 subjects, including 11,986 with ASD, was published [12]. By describing rare de novo and inherited coding variations, it identified 102 putative ASD risk genes at a false discovery rate of ≤0.1. Thirty of these genes were identified for the first time, as they had never been significantly enriched for de novo and/or rare variants in previous studies. Notwithstanding these efforts, and the important knowledge arising from this recent exome sequencing study, no major causative gene has been isolated so far [13,14]. Consequently, several essential questions regarding the pathophysiology of ASD, including which cell types and biological processes are disrupted, are still poorly understood. Some advancements in the detection of rare genetic variants have been made by studying genetic syndromes related to ASD [9], and which have shed a new light on the pathophysiology of the disorder [15]. By studying the syndromic ASD, some evidence has emerged that even heterogeneous ASD phenotypes might possibly present with a convergent pathophysiology [8].

Considering this complex challenge, network-based analyses have been suggested as possible powerful tools to discover interaction patterns in ASD pathophysiology, and to provide a functional explanation of genetic heterogeneity in ASD [16,17]. Several authors have investigated possible methods of producing network-based prioritization of ASD genes [16]. In particular, machine learning algorithms have been employed in order to delineate the ASD architecture [18,19,20]. In addition, cluster analysis has seen increasing applications in biomedicine, to further the understanding of ASD [21]. In order to better assess the strength of evidence associated with ASD candidate genes [11] large systematic databases are needed. There are already several databases collecting ASD variant data, for example SFARI [22], AutismKB [23], and the one developed within the Autism Speaks’ MSSNG project [24]. However, even though advances in technology, and widespread databases, are continuing to disentangle the genetic aspects of ASD, the pathogenesis of the disorder is still a matter of speculation.

Moving from this background, the present study aimed to look at gene-networks rather than at specific gene variations, and to identify and categorize those biological processes that might act as a possible pathophysiological substrate for ASD. Hence, instead of focusing on single genes, DNA segments, or genetic variants, we prioritized those biological processes that occur in genetically related clusters of patients. The recent availability of VariCarta [25], an ASD specific database, allowed executing a preliminary analysis on an adequate amount of individuals. The VariCarta dataset is freely available at https://varicarta.msl.ubc.ca/index, both using a web interface or by downloading the whole dataset in csv format. All data generated or analyzed during the present study are included in this published article and its additional files.

We first defined a metric to measure the genetic similarity between patients according to their variations, and then applied hierarchical clustering in order to identify groups of genetically similar individuals [26]. Afterwards, we proceeded with the enrichment analysis upon each cluster of patients, so as to identify ASD-related biological pathways which might have been impacted by the genetic variations. Finally, we discuss the implications of biological process knowledge for clinical practice.

## 2. Materials and Methods

### 2.1. Gene Database

For this research work we used the VariCarta dataset from British Columbia University, a web-based database of human DNA genetic variants identified in individuals with an ASD diagnosis [25]. It also presents a list of genes, with both a set of variants, and the reference to the individual affected by the variant. This information was fundamental for the cluster analysis we carried out. However, no further details about individuals, e.g., age, gender, ethnicity, family relationship, as well as information regarding the homozygous/heterozygous status of each subject, are reported in the database. VariCarta was developed to support the identification of genetic variants in individuals with ASD. To address this challenge, it was necessary to collect a large quantity of subject information, through the aggregation of data, with the risk of methodological inconsistencies and subject overlap across studies. The VariCarta developers tried to tackle this task by collecting and cataloging literature-derived genomic variants found in ASD subjects, through the use of ongoing semi-manual curation, and with a robust data import pipeline. Thus, while developing the database, it was possible to find and correct errors, to convert variants into a standardized format, to identify and harmonize cohort overlaps, and to document data origin. The database is constantly updated with new relevant gene-targeted ASD research papers. The version analyzed in this study, dated 12 November 2019, contains 184,212 variant events from 13,069 individuals, collected across 69 publications. It consists of 211,669 records, each one containing a genetic variant event, as reported in the paper from which it was retrieved, irrespective of whether it is considered (according to today’s knowledge) a variant that contributes to ASD or not. However, several of the 184,212 events are duplicated variants, as they have been retrieved from different publications that included the same individual. The actual number of unique variants is 149,695. All the variants included in VariCarta are the result of previously published controlled studies, which had suggested an association between the variant and ASD. The variants belonging to controls have not been included in VariCarta [25]. Variants related to whole-genome sequencing have been retrieved from 2062 individuals. Since a single variant event can be reported by more than one paper, duplicated events were removed during the analysis. Most of the variants retrieved from the literature, and included in VariCarta, are de novo variants; considering whole-genome sequencing, de novo variants cover 99.7% (149,270/149,695) of all variants, and only 0.3% are inherited variants.

In VariCarta, an exome aggregation consortium minor allele frequency (ExAC) [27] is reported for 0.9% of unique variants of the whole-genome sequencing (1290/149,695). Out of the 1290 variants reporting an ExAC score, only 0.7% (9/1290) have an ExAC score above 0.05, and 2.5% (32/1290) an ExAC above 0.01. Therefore, the large majority of the variants reported by VariCarta are non-common. A combined annotation dependent depletion score (CADD) [28], is reported for 1.3% of unique variants (1945/149,695). For all these variants, we also used ExAC and CADD scores to filter gene lists before the enrichment analysis.

### 2.2. Analysis of the Dataset

The dataset is accessible both using a web interface and by downloading the whole dataset in csv format. As the web interface allows limited research, we downloaded the whole dataset in csv format. Each row of the dataset is a variant event, including, among others, the symbol of the affected gene, the category of variant (synonymous SNV and nonsynonymous SNV, frameshift insertion, etc.), the adopted sequencing type (whole genome sequencing, exome sequencing, targeted sequencing), and the subject id, which is a unique identifier of the patient presenting the variant. In VariCarta the amount of synonymous (SNV) variants is limited: considering whole-genome sequencing, there are only 556 synonymous (SNV) variants among the 149,695 unique variants (i.e., 0.4%). The dataset is also provided with reference information, which allows tracing the paper where the information was gathered from. In the current study we focused on whole genome sequencing, and did not consider targeted analysis sequencing, because this technique is focused on identifying specific genes highly related to a disease, assuming that these are known. Such an assumption can be made only for well documented causes of ASD, such as tuberous sclerosis or Fragile X syndrome, diagnosed in about 15% of individuals with ASD [29]. This is not the case for “idiopathic ASD”, which represents the majority of all ASD diagnoses. Therefore, limiting the analysis to some genes, while ignoring others, could lead to a limited detection of gene variants in a single subject. Furthermore, in the database the number of records revealed by targeted sequencing was only about 2% of all records, (3698/184,212 records). From the remaining records we created a subset of unique variants, composed of 81.3% of all records collected in the dataset (149,695/184,212). We selected two features, “Gene Symbol” and “Subject id”, and made a pairwise comparison of the patients of each group to build a “patient similarity matrix”, defined as follows: Let A be a matrix N × N where N is the number of patients; let G_i_ be the set of genes affected by a variation in the subject i, and G_j_ the set of genes affected by a variation in the subject j; we defined each element a_i,j_ of A as the intersection between G_i_ and G_j_. A Log2 transformation was applied to each element a_i,j_ of the matrix A in order to normalize the results.

In order to sort rows and columns, to highlight clusters underlying the common occurrence of gene networks in patients, we used the clustermap function of the Python library Seaborn [30]. This function leverages hierarchical clustering [31] and heatmaps [32] to identify clusters inside the rows and columns of the input matrix, which can be either a rectangular observation matrix or a square distance matrix. Hierarchical clustering is a widely used clustering algorithm that is able to identify hierarchical relations between groups [33]. It is particularly effective when some hierarchical structure (like a taxonomy) is expected to be identified, and when the number and nature of groups and subgroups are not known in advance. Hierarchical clustering has been used in biology [34] since the 70s. Today it is applied to genetics, combined with heatmaps for microarray analysis [35], and recently, it has been applied to psychiatry to identify subgroups of patients with ASD, based on comorbidity [36] or phenotype analysis [37,38].

### 2.3. Hierarchical Agglomerative Clustering (HAC)

The specific algorithm we used was hierarchical agglomerative clustering (HAC) [39], a bottom-up iterative process. It tries to progressively identify sets of similar subsets of data, by leveraging a distance function, also called hierarchical linkage. In the HAC process items are iteratively aggregated using the distance function to evaluate similarities between subgroups.

The HAC we used during the analysis was the Python implementation, provided by the Seaborn library in its clustermap function. Seaborn’s clustermap combines HAC, to sort rows and columns of its heatmaps, with showing dendgrograms on the axes, to highlight the hierarchical structure. The implementation of HAC included in clustermap begins with a forest of clusters that have yet to be used in the hierarchy being formed. When two clusters B and C from the forest are combined into a single BC cluster, B and C are removed from the forest, and BC is added to the forest. When only one cluster remains in the forest, the algorithm stops, and this cluster becomes the root. A distance matrix is maintained at each iteration. The *d*[*i,j*] entry corresponds to the distance between clusters *i* and *j* in the original forest. At each iteration, the algorithm has to update the distance matrix to reflect the distance between the newly formed cluster, BC, and the remaining clusters in the forest. The computation of the distance, *d*[*i,j*], depends on the method used. In the event, we adopted the “complete” method that applies the following function:(1)d(u,v)=max(dist(u[i],v[j]))

This function, also called the farthest point algorithm, or Voorhees algorithm, implies that the distance between two clusters, *u* and *v*, is the maximum distance between the farthest points. We applied this method in order to try to maximize the differences between clusters. The function used to measure the distance between the points was the Euclidean distance (2-norm). We then used the partitions to try to identify gene networks that could be characteristic of each subgroup.

### 2.4. Enrichment Analysis

In order to identify biological processes that could be impacted by the gene variations of each subgroup we used enrichment analysis. Gene set enrichment analysis [40] (GSEA) (also called functional enrichment analysis) is a method of identifying classes of genes or proteins that are over-represented in a large set of genes or proteins, and may be associated with disease phenotypes. The method uses statistical approaches to identify significantly enriched or depleted groups of genes. This can be done by comparing the input gene set to each of the bins (terms) in the gene ontology. A statistical test can be performed for each bin to evaluate if it is enriched for the input genes. The results for each pathway are expressed in terms of fold enrichment (FE), i.e., the ratio between the number of genes present in the cluster list belonging to that pathway, and the number of genes expected to belong to that pathway in a random set of genes of the same size. The setting used for the enrichment analysis was the GO Ontology database (Released 21 February 2020) [41]. The applied reference list of expected genes was the one for homo sapiens.

### 2.5. Statistical Analysis

PantherDB [42] was applied for the Enrichment analysis (PANTHER Overrepresentation Test; Released 7 April 2020). To verify the statistical significance of the submitted set of genes, we applied Fisher’s exact test. The obtained raw *p*-values were adjusted for multiple comparisons by means of the false discovery rate method (FDR) [43]. A conservative statistical significance threshold of *p* < 0.005 (two-tailed) was applied for all analyses. As both raw and FDR-adjusted *p*-values are strongly dependent on sample size, once the statistically significant terms were identified, we ranked the biological processes by fold enrichment, a measure of the effect-size [44]. In order to highlight interactions between the set of identified genes we used String DB [45] “Gene Network View”. In the produced network interaction graph, the number of edges represents the existing interactions between the submitted genes. This was compared with the expected number of edges, i.e., the expected interactions between a set of random genes of the same size. STRING DB was also used to compute the protein–protein interaction (PPI) enrichment *p*-values on partitions identified by the intersection of gene sets identified by the clustering.

## 3. Results

### 3.1. Cluster Heatmap

The cluster heatmap of the dissimilarity matrix of the patients is presented in Figure 1a. All 2062 patients are distributed along the two axes. Each number is the subject identification (id) assigned to a specific patient as it is presented in the VariCarta database. Each cell of the matrix is the log2 transformation of the number of the common gene variations between two patients. The usual diagonal line that would show the comparison of each patient with himself was suppressed by the clustermap function.

Further analyses were performed on the macro-clusters represented in Figure 1b. The dendrogram showing their relationships was extracted from the one visible in the cluster heatmap axes, limiting it to the third hierarchy level.

The whole-genome matrix presented a high number of shared variations between patients. Three clusters were identified (A, B, and C). We labeled the three clusters according to their density, i.e., the amount of shared variations between the patients, from the highest to the lowest. We considered as a density factor (DF) the average value of all the elements of a cluster. Each value of the matrix is the log2 transformation of the number of gene variations in common between the two patients representing, respectively, the row and the column of the matrix. Cluster A is composed of 574 patients (27.8% of the considered individuals), sharing 8357 gene variations (DF: 1.429); Cluster B is composed of 507 patients (24.6%), sharing 6818 gene variations (DF: 1.014); Cluster C is composed of 650 patients (31.5%), sharing 7704 gene variations (DF: 0.720). Beyond the three clusters, we also identified 331 patients who shared no, or very few, gene variations (all together 41 genes). We included them in the “mixed group” D, whose DF was just 0.001.

In order to estimate the possible impact of inherited variants in computing similarity between patients, we also computed a similarity matrix based only on inherited variants (0.3% of all variants). The result showed very small clusters, with leak similarity values, between patients. Subsequently, we computed the same clustering, considering only the de novo variants (99.7% of all variants), yielding the same results in terms of cluster composition as the ones presented in Figure 1b. Therefore, the few inherited variants were not excluded from the subsequent enrichment analysis.

As the VariCarta database is made up of a collection of various datasets gathered from published studies, we checked if the identified clusters overlapped with the populations of these datasets. For each cluster (A, B, C, and D), we identified the published papers that contained individuals included in the cluster. The list of papers by cluster showed that there was no cluster whose original dataset was unique or originating only from a few papers [Appendix A]. We could therefore exclude overlapping between the original datasets and the identified clusters.

### 3.2. Enrichment Analysis

For each of the clusters shown in Figure 1b we extracted a list of common gene variations, gene variations in at least two patients of the cluster, and then executed an enrichment analysis. The results of the 40 biological processes with the highest FE values are presented in Table 1. Only clusters A, B, and C were considered, as the mixed group, D, did not return any statistically significant results. The involved biological processes were ranked by fold enrichment to highlight the most specific processes involved.

### 3.3. Cluster Comparisons

Most of the processes of the three clusters are common, even though some differences can be detected, either because of the different order in the list, which indicates a different weight of that process in the cluster, or because of the presence of cluster-specific processes. For example, as shown in Table 1, in clusters A and B the majority of the biological processes with the highest FE relate to neuron and synapse activity, while the biological processes of cluster C are more shifted towards organ development and morphogenesis.

For each cluster we then extracted variants shared by at least two subjects in the same cluster. As shown in Figure 2, by intersecting the three clusters we identified seven partitions. We executed enrichment analysis on all the partitions separately. None of the partitions, except the intersection between the three clusters returned significant results. Table 2 shows the enrichment analysis on the ABC intersection.

The resulting network interaction graph, generated by String-DB, is presented in Figure 3. For the sake of clarity, the analysis was limited to genes, which were present in at least 50 patients. The number of nodes of the network was 316, and the number of edges 990. The expected number of edges was 293; 3.4 times less than the number of edges of the submitted network.

### 3.4. Enrichment Analysis on ExAC Filtered Variants

We then filtered the dataset by allele frequency, using ExAC for the cases in which this value was reported. The analysis on each distinct cluster did not return any significant results. An additional enrichment analysis was performed on the set of genes belonging to the intersection between clusters A, B, and C. First, we executed the analysis considering only variants with an ExAC score < 0.05, and then we restricted the list of genes, considering only variants having an ExAC score < 0.01.

In both cases we identified three biological processes yielding an FE ≥ 12 and an FDR < 0.005: “observational learning” (GO:0098597), “imitative learning” (GO:0098596), and “vocal learning” (GO:0042297). The next processes, ordered by FE values, were processes related to neurotransmission, like “membrane depolarization during action potential” (GO:0086010), “dendrite morphogenesis” (GO:0048813), and “regulation of NMDA receptor activity” (GO:2000310). Further details are reported in the Appendix A.

### 3.5. Enrichment Analysis on CADD Filtered Variants

In order to perform the enrichment analysis by filtering genes in terms of CADD, we divided the variants related to the intersection between clusters A, B, and C into five groups, according to their CADD score: <10, 10–19, 20–29, 30–39, ≥40. We then executed an enrichment analysis for each subgroup. In all subgroups, biological processes with the highest FE and FDR < 0.005 were related to neurotransmission. In particular, in the subgroup where the deleteriousness of variants was higher (≥40), we identified a set of biological processes that were all related to synaptic transmission, namely: “negative regulation of excitatory postsynaptic potential” (GO:0090394), with FE 43.58; “presynaptic membrane assembly” (GO:0097105), with FE 43.58; “presynaptic membrane organization” (GO:0097090), with FE 39.94; “prepulse inhibition” (GO:0060134), with FE 31.96; “modulation of excitatory postsynaptic potential” (GO:0098815), with FE 18.64; “membrane assembly” (GO:0071709), with FE 17.12; “negative regulation of synaptic transmission” (GO:0050805), with FE 12.91; “regulation of postsynapse organization” (GO:0099175), with FE 12.69. The complete list of biological processes and the FE of each subgroup is reported in the Appendix A.

## 4. Discussion

### 4.1. Enrichment Analysis

We identified three main genetic clusters of ASD patients, each one characterized by a set of genetic variants. The subsequent enrichment analysis (EA), performed upon the clusters’ genes, allowed pinpointing biological processes, both common to the three clusters, and cluster-specific. Most of the processes that were common to the three clusters were involved in neuron projection guidance and morphogenesis. Proper plasticity of axon and dendrites is a highly dynamic process leading to functional circuit formation [46], which is crucial in the very early stages of brain development [47,48]. These processes involve numerous ASD related genes associated with disrupted synaptic connectivity and function [12,49]. Neuronal migration and morphogenesis defects in ASD contribute to an altered cortical connectivity in different brain regions [50,51], such as the well-known prefrontal area [52]. Therefore, in the developing brain, a premature alteration in neuronal plasticity, affecting mostly cortical regions involved in cognitive and behavioral functions, might trigger the autistic phenotype [53,54]. Furthermore, fMRI [55,56] and electrophysiological [57,58] studies have also emphasized the role of an atypical interconnection between specific brain areas in ASD. Considering these findings, the biological processes we underlined for each cluster give further support to the theory that neurogenesis and its molecular microenvironment are associated with autism.

### 4.2. Cluster Comparisons

This pathogenic background was also indicated by the enrichment analysis performed on the intersection of the three clusters, which confirmed the role in ASD etiopathology of gene networks related to biological processes affecting neuron development, for example “axon guidance”, “neuron projection guidance”, or “dendrite morphogenesis”. As further evidence of disrupted connectivity in ASD, the EA on the single clusters, and on their intersection, showed the importance of processes like “cell junction assembly” and “synapse assembly”. Moreover, when filtering genes by CADD, the highest fold enrichment values were associated with dendrite and synaptic related biological processes. The integrity of synaptic proteins and cell adhesion molecules is crucial for synaptic formation, signal transduction, and transmission [59]. Thus, synaptic dysfunction due to molecular damage, and a consequently altered signal conduction, are consistent with ASD etiopathology [60]. Overall, the presence of shared genetic networks between the three clusters and, consequently, of biological processes involving both neurite and synaptic formation and function might be the underlying cause of comparable ASD phenotypes among different individuals.

Interestingly, when filtering genes by ExAC scores, the analysis performed on the intersection between the clusters identified significant processes connected to observational, imitative, and vocal learning. Actually, imitation impairments in patients with ASD are typically related to their deficits in connecting and learning from others [61], and the comprehension of their interpersonal synchrony skills is now fundamental to trace the adaptive functioning and, consequently, the ASD severity, in children [62].

Besides the shared genetic networks, our enrichment analysis also signaled some processes that differentiate the three clusters, and that could possibly cause phenotypic dissimilarities. Actually, the analysis brought to light the presence of cluster-specific processes, which were not shared with the other clusters; for example, in the case of the biological process “kidney development” in Cluster A, or “cardiac septum development” in Cluster C. Although this observation should be considered with caution, as the relative FDR *p*-values, though statistically significant, are close to the threshold of <0.005, it is noteworthy that recently ASD-related processes regarding the heart and kidney have become features of scientific and clinical interest [63,64]. Both cluster-specific and common atypical biological processes might interact to shape the specific phenotype of each individual. At any rate, these observations point to the fact that when considering the disrupted processes of the different ASD phenotypes we should not only look at the central nervous system but at other systems as well. Thus, this is an important piece of evidence supporting the polygenic model that assumes that ASD is the result of rare and common variant combination [65].

In our study, we also identified a group of 331 people composed of individual patients that had either no or just a few variations in common with the rest of the sample. While individuals belonging to clusters A, B, and C show substantial genetic similarities between them, the individuals of this last group seem to be genetically uncorrelated. It might therefore be speculated that, in relation to ASD development, genetics might play a stronger role in individuals of clusters, A, B, and C than in those outside these clusters. These findings further support an ASD heterogeneity, as they may indicate that the clinical phenotype might also be the outcome of common genetic variants that could interact with main variant, epigenetic [65], or environmental factors [6], which have not been considered in this pilot study.

### 4.3. Translation into Clinical Research

An element of this preliminary work was the use of patient similarity analytics. This data-driven approach has already been applied in medicine to the study of different diseases, including behavioral disorders [66] and other psychiatric diseases [67]. Specifically, in ASD research and clinical medicine patient similarity algorithms could play a role in structuring autism heterogeneity, i.e., in identifying ASD patient subgroups who share the same etiopathology. Subgroups of patients can then be assessed by additional fine-grained stratifications, based, for example, on their genetic characteristics or biological processes. Once subsets of patients have been isolated and characterized, it becomes possible to perform systematic individualized analyses, by evaluating the distance of a single patient from each subgroup. Enumerating both the common and specific processes that characterize different ASD subpopulations might allow understanding which, among a large number of autism genetic variants, are those that play a major role in the etiopathology of ASD. This could help to bridge the crucial gap between the detection of new genetic risk variants for ASD [68], and the clinical translation of these discoveries. The presented methodology might be a useful starting point developing more focused research on subgroups of individuals with ASD, by singling out homogeneous patient cohorts. Such an approach could help to shed light on the relative impact of specific biological processes among different autistic phenotypes. Although patient similarity analytics is still in development, it promises to be helpful in identifying the contribution of noncoding mutations to autism risk [69], in predicting patients’ trajectory over time [70], in providing clinical decision support [71], and in tailoring individual treatments [72].

### 4.4. Limitations

Our preliminary findings should be viewed with several limitations in mind. First, as the VariCarta database we used contains neither genetic data of family members of the ASD patients, nor of neurotypical subjects, we could not perform the same analysis on a group of non-affected individuals. This is an important limitation of the database, as we know that comparing family-based cases with unaffected siblings is crucial in disentangling de novo from inherited variants [12]. Similarly, an analysis based on population stratification was not possible, since no specifics about individuals (as age, gender, ethnicity, family relationship, etc.) are reported in VariCarta. Particularly, VariCarta does not provide information regarding the homozygous/heterozygous status of each individual. Therefore, the variability of the effects of the variants related to this aspect could not be assessed in the present analysis, even though, considering the design of this study, the lack of this information would have had only a minor impact on the overall results. Additionally, ExAC or CADD scores are reported in VariCarta only for a limited number of variants. Therefore, we could not remove common variants, and evaluate the deleteriousness of variants in the cluster analysis, since too few variants would have been available for a significant clustering. However, we deem it a minor limitation, as most of the existing studies on autism genetics retrieved by VariCarta report mainly rare variants [25]. Furthermore, our analyses on the subset of variants, where these scores were reported, confirmed the overall results. It also has to be stressed that we analyzed a database which did not include epigenetic or environmental factors, thus limiting the characterization of the subgroups only to genetic variant events. Second, we included in the analysis all variations, without differentiating their type (base substitution, deletion, or insertions), category of nucleotide variation, or sequence variation (exonic, intronic). Of course, different types of genetic variants may have completely different functional effects; some might strongly contribute to the development of ASD, others might not. Actually, most of the genetic variants may not be involved in ASD at all, even if the variants included in VariCarta have been retrieved from published controlled studies, where the variant had been associated with ASD. Third, the absence in the dataset of the description of each phenotype, including gender and IQ, did not allow us to confirm our clustering results. Although there is evidence that biological and behavioral characteristics may differ between males and females [50], in this study we did not have the tools to delve into this aspect. Moreover, we could not verify if the biological processes impacted by the variant networks of each cluster actually produced different phenotypes. Furthermore, 30–50% of ASD individuals present with comorbid intellectual disability, or with other neurodevelopmental disorders [15], and family studies have shown high heritability estimates in ASD, but lower ones with respect to other neurodevelopmental disorders [73]. This lack of knowledge of the individual phenotype did not allow distinguishing between variations that influence ASD, and variations that influence other neurodevelopmental disorders.

## 5. Conclusions

Given the preliminary characteristics of our study, it will be important for future research to replicate our findings, comparing them with genetic data of neurotypical people. A comparison of the identified biological processes with associated phenotypes will definitely be necessary to confirm the clinical validity and usefulness of our results. To improve our understanding of ASD heterogeneity, neurobiology, and genomic architecture, study designs have to increasingly consider innovative methodologies and newly developed biomedical informatics. Integrated genomic approaches, supported by advanced mathematical modeling able to also contemplate environmental factors, might lead to a better comprehension of the etiology and the pathogenic mechanisms of ASD.

## Figures and Tables

**Figure 1 genes-11-01476-f001:**
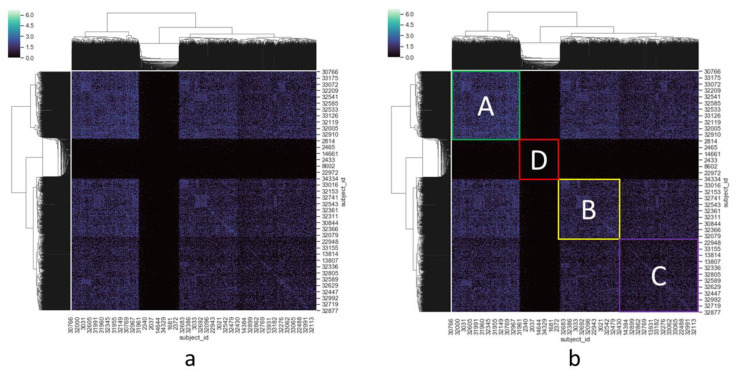
(**a**) Both axes are represented by patients. For the sake of readability, only 2% of the 2062 subject ids are represented in the heatmap. Each value of the matrix is the result of a pairwise comparison of patients, computed by calculating the log2 of the size of the intersection between the sets of gene variations in the pair of patients. The values range from 0 (no gene variations in common) to 6 (more than 60 gene variations in common). The dendrograms show the cluster hierarchy proposed by the HAC algorithm. Considering the first three hierarchical levels, four macro-areas, with different densities, can be identified on the diagonal axis. (**b**) The four areas are highlighted in different colors. A, B, and C represent three clusters. Each cluster is characterized by a different density, expressed by density factor (DF), i.e., the mean value of all the cells belonging to the cluster. Cluster A: DF = 1.429; Cluster B: DF = 1.014; Cluster C: DF = 0.720. Beyond the three clusters, patients who shared no, or very few, gene variations were also identified, and included in the “mixed group” (D): DF = 0.001. Non-highlighted areas represent overlapping between clusters, implying the existence of variants in common to all subgroups.

**Figure 2 genes-11-01476-f002:**
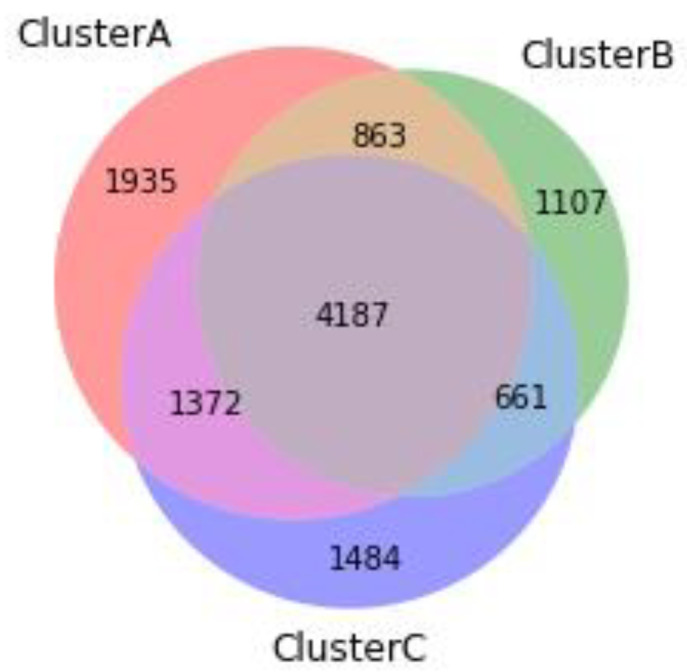
Venn diagram of gene overlap among clusters A, B, and C. By intersecting clusters A, B, and C, seven partitions can be identified. Figures overwritten on the seven partitions of the diagram show the number of genetic variants shared by at least two subjects of each intersection. Overall, 11,609 genetic variants were isolated: 4187 of these variants (36.1%) were common to the three clusters; 1935 (16.7%), 1107 (9.5%), and 1484 (12.8%) variants were specific to clusters A, B, and C, respectively. The subsequent enrichment analysis was executed on each partition separately. Only the intersection between the three clusters returned a significant result: the FDR protein–protein interaction (PPI) enrichment *p*-value for the intersection of clusters A, B, C was *p* < 1.0 × 10^−16^.

**Figure 3 genes-11-01476-f003:**
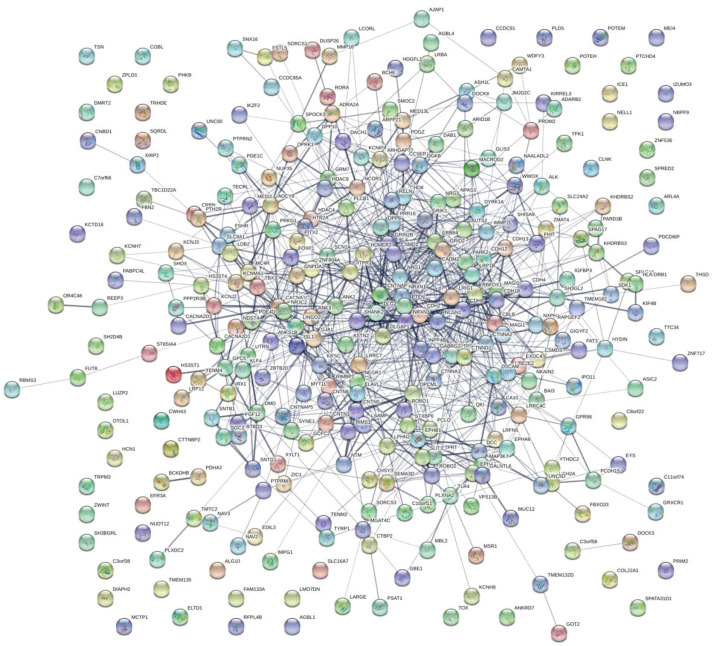
Network interaction graph of the intersection between clusters A, B, and C. Analysis of genes present in at least 50 patients. Nodes: 360. Edges: 990. Expected number of edges: 293. The number of edges represents the existing interactions between the submitted genes. This was compared with the expected number of edges, i.e., the expected interactions between a set of random genes of the same size.

**Table 1 genes-11-01476-t001:** Enrichment analysis on Clusters A, B, and C, ranked by fold enrichment. Whole-genome sequencing data.

Cluster A			Cluster B			Cluster C		
GO Biological Process	FE	FDR	GO Biological Process	FE	FDR	GO Biological Process	FE	FDR
neuron projection guidance (GO:0097485)	1.75	6.88 × 10^−6^	dendrite morphogenesis (GO:0048813)	2.38	3.03 × 10^−3^	neuron recognition (GO:0008038)	2.46	2.19 × 10^−3^
axon guidance (GO:0007411)	1.74	9.84 × 10^−6^	neuron projection guidance (GO:0097485)	2.07	7.85 × 10^−10^	ventricular septum development (GO:0003281)	2.18	2.85 × 10^−3^
regulation of axonogenesis (GO:0050770)	1.71	9.35 × 10^−4^	axon guidance (GO:0007411)	2.07	1.29 × 10^−9^	cardiac septum development (GO:0003279)	2.01	5.15 × 10^−4^
axonogenesis (GO:0007409)	1.68	3.30 × 10^−7^	synapse assembly (GO:0007416)	2.07	1.28 × 10^−3^	negative regulation of developmental growth (GO:0048640)	1.89	4.10 × 10^−3^
neuron projection morphogenesis (GO:0048812)	1.68	2.13 × 10^−9^	action potential (GO:0001508)	2.06	2.60 × 10^−3^	cell–cell junction assembly (GO:0007043)	1.87	3.43 × 10^−3^
plasma membrane bounded cell projection morphogenesis (GO:0120039)	1.68	1.95 × 10^−9^	developmental growth involved in morphogenesis (GO:0060560)	2.05	4.71 × 10^−4^	neuron projection morphogenesis (GO:0048812)	1.86	1.45 × 10^−13^
regulation of JNK cascade (GO:0046328)	1.68	2.23 × 10^−3^	regulation of synapse assembly (GO:0051963)	2.02	1.89 × 10^−3^	cell morphogenesis involved in neuron differentiation (GO:0048667)	1.85	5.59 × 10^−12^
cell morphogenesis involved in neuron differentiation (GO:0048667)	1.68	3.49 × 10^−8^	cell morphogenesis involved in neuron differentiation (GO:0048667)	2.00	1.31 × 10^−14^	neuron projection guidance (GO:0097485)	1.84	5.84 × 10^−7^
cell projection morphogenesis (GO:0048858)	1.67	2.25 × 10^−9^	axonogenesis (GO:0007409)	2.00	2.11 × 10^−12^	plasma membrane bounded cell projection morphogenesis (GO:0120039)	1.84	3.30 × 10^−13^
regulation of cell junction assembly (GO:1901888)	1.67	1.34 × 10^−3^	negative regulation of cell morphogenesis involved in differentiation (GO:0010771)	2.00	4.30 × 10^−3^	axon guidance (GO:0007411)	1.84	6.70 × 10^−7^
axon development (GO:0061564)	1.67	1.76 × 10^−7^	neuron projection morphogenesis (GO:0048812)	2.00	3.64 × 10^−16^	axonogenesis (GO:0007409)	1.84	6.78 × 10^−10^
cell part morphogenesis (GO:0032990)	1.66	2.93 × 10^−9^	plasma membrane bounded cell projection morphogenesis (GO:0120039)	1.99	4.61 × 10^−16^	cell projection morphogenesis (GO:0048858)	1.83	4.04 × 10^−13^
renal system development (GO:0072001)	1.62	1.87 × 10^−4^	regulation of cell junction assembly (GO:1901888)	1.99	2.52 × 10^−6^	axon development (GO:0061564)	1.82	1.99 × 10^−10^
telencephalon development (GO:0021537)	1.62	4.40 × 10^−4^	cell projection morphogenesis (GO:0048858)	1.97	1.10 × 10^−15^	regulation of axonogenesis (GO:0050770)	1.82	1.27 × 10^−4^
kidney development (GO:0001822)	1.62	3.42 × 10^−4^	cell junction assembly (GO:0034329)	1.95	7.79 × 10^−8^	cell part morphogenesis (GO:0032990)	1.80	8.91 × 10^−13^
regulation of small GTPase mediated signal transduction (GO:0051056)	1.61	6.55 × 10^−5^	cell-cell junction assembly (GO:0007043)	1.93	3.16 × 10^−3^	regulation of small GTPase mediated signal transduction (GO:0051056)	1.78	3.01 × 10^−7^
regulation of neuron projection development (GO:0010975)	1.61	5.54 × 10^−8^	synapse organization (GO:0050808)	1.93	2.32 × 10^−8^	cell junction assembly (GO:0034329)	1.76	8.13 × 10^−6^
regulation of cell morphogenesis involved in differentiation (GO:0010769)	1.60	1.26 × 10^−4^	axon development (GO:0061564)	1.92	6.41 × 10^−12^	cell–cell junction organization (GO:0045216)	1.76	1.54 × 10^−3^
positive regulation of neuron differentiation (GO:0045666)	1.60	1.03 × 10^−5^	cell part morphogenesis (GO:0032990)	1.92	5.66 × 10^−15^	neuron projection development (GO:0031175)	1.75	8.64 × 10^−15^
cell morphogenesis involved in differentiation (GO:0000904)	1.60	1.97 × 10^−8^	cell morphogenesis involved in differentiation (GO:0000904)	1.90	1.35 × 10^−15^	regulation of cell junction assembly (GO:1901888)	1.73	5.78 × 10^−4^

FE: Fold Enrichment; FDR: False Discovery Rate *p*-value. Biological processes are identified by their reference numbers (GO:XXXXXX) in the Gene Ontology. The first 40 FE ranked biological processes of each cluster are shown. An additional table file shows the full list of the 58 biological processes of Cluster A, the 135 processes of Cluster B, and the 87 processes of Cluster C, with a FE ≥ 1.5 [Appendix A]. The occurrences of gene variation are listed in the Appendix A.

**Table 2 genes-11-01476-t002:** Enrichment analysis on the intersection among clusters A, B, and C. Whole-genome sequencing data.

GO Biological Process	FE	FDR
cell-cell adhesion mediated by cadherin (GO:0044331)	3.85	1.74 × 10^−3^
outflow tract septum morphogenesis (GO:0003148)	3.62	4.27 × 10^−3^
synaptic transmission, glutamatergic (GO:0035249)	3.36	8.03 × 10^−4^
neuron recognition (GO:0008038)	3.30	1.50 × 10^−4^
glutamate receptor signaling pathway (GO:0007215)	3.18	4.96 × 10^−4^
dendrite morphogenesis (GO:0048813)	3.05	1.98 × 10^−4^
receptor localization to synapse (GO:0097120)	3.05	2.83 × 10^−3^
heterophilic cell–cell adhesion via plasma membrane cell adhesion molecules (GO:0007157)	2.99	2.58 × 10^−3^
regulation of cell–substrate junction organization (GO:0150116)	2.87	2.72 × 10^−4^
synapse assembly (GO:0007416)	2.84	2.45 × 10^−6^
heart valve morphogenesis (GO:0003179)	2.82	2.78 × 10^−3^
regulation of focal adhesion assembly (GO:0051893)	2.81	6.54 × 10^−4^
regulation of cell–substrate junction assembly (GO:0090109)	2.81	6.52 × 10^−4^
retina morphogenesis in camera-type eye (GO:0060042)	2.79	1.71 × 10^−3^
neuron projection guidance (GO:0097485)	2.75	2.15 × 10^−15^
axon guidance (GO:0007411)	2.73	5.03 × 10^−15^
negative regulation of axonogenesis (GO:0050771)	2.67	8.24 × 10^−4^
adherens junction organization (GO:0034332)	2.66	2.70 × 10^−3^
regulation of glutamate receptor signaling pathway (GO:1900449)	2.64	1.56 × 10^−3^
cardiac septum morphogenesis (GO:0060411)	2.64	6.87 × 10^−4^
negative regulation of cell morphogenesis involved in differentiation (GO:0010771)	2.59	6.52 × 10^−5^
positive regulation of synapse assembly (GO:0051965)	2.59	2.49 × 10^−3^
protein localization to synapse (GO:0035418)	2.56	4.75 × 10^−3^
regulation of neurotransmitter receptor activity (GO:0099601)	2.52	1.13 × 10^−3^
mechanoreceptor differentiation (GO:0042490)	2.52	4.82 × 10^−3^
regulation of synaptic transmission, glutamatergic (GO:0051966)	2.50	3.75 × 10^−3^
cell morphogenesis involved in neuron differentiation (GO:0048667)	2.48	8.42 × 10^−20^
neuron projection morphogenesis (GO:0048812)	2.47	1.37 × 10^−21^
axonogenesis (GO:0007409)	2.47	1.72 × 10^−16^
regulation of synapse assembly (GO:0051963)	2.46	1.41 × 10^−4^
plasma membrane bounded cell projection morphogenesis (GO:0120039)	2.45	2.14 × 10^−21^
cell projection morphogenesis (GO:0048858)	2.43	4.51 × 10^−21^
positive regulation of phosphatidylinositol 3-kinase signaling (GO:0014068)	2.43	1.35 × 10^−3^
dendrite development (GO:0016358)	2.42	1.01 × 10^−4^
neural crest cell differentiation (GO:0014033)	2.42	1.85 × 10^−3^
regulation of sodium ion transport (GO:0002028)	2.39	2.01 × 10^−3^
axon development (GO:0061564)	2.38	2.47 × 10^−16^
regulation of potassium ion transmembrane transport (GO:1901379)	2.38	3.48 × 10^−3^
cell part morphogenesis (GO:0032990)	2.38	1.50 × 10^−20^
regulation of potassium ion transport (GO:0043266)	2.37	6.70 × 10^−4^

FE: Fold Enrichment (FE); FDR: False Discovery Rate *p*-value. Biological processes are identified by their reference numbers (GO:XXXXXX) in the Gene Ontology. The first 40 FE ranked biological processes related to the intersection among clusters A, B, and C are reported. An additional table file shows the full list of the 359 biological processes with a FE ≥ 1.5 [Appendix A]. The occurrences of gene variations are listed in the Appendix A.

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
