# Peer review of "Clustering Analysis Supports the Detection of Biological Processes Related to Autism Spectrum Disorder"

_genes, 2020, doi:10.3390/genes11121476_

Round 1

Reviewer 1 Report

In this manuscript Gialloreti et al has studied the genes reported to be associated with ASD in the previously published literatures, on the basis of their clustering. Clustering approaches are prenominal use on the basis of their phenotypic. Thus, the authors bring novelty in their work. However, there are many limitations min this study. Interestingly, authors have mentioned majority of the limitation in the manuscript itself. Mode of inheritance is one of my major concern for this study which is key limitation. Since the authors is considering genes observed in 2 patients for their analysis and if such genes are from close/ distant relatives; then there will be bias in the findings. Additionally, I have following suggestions which will made this paper interesting to autism researchers.

  1. Can you divide the genes into groups on the basis of CADD (<10, 10-20, 20-30, >40) and do the same analysis.?
  2. Do the same for allele frequency <0.01 vs <0.05.
  3. Similarly divide the genes based on the pattern of inheritance de novo vs inherited?

If the findings don’t change with these factors; then it will be interesting to the researchers.

Author Response

In this manuscript Gialloreti et al has studied the genes reported to be associated with ASD in the previously published literatures, on the basis of their clustering. Clustering approaches are prenominal use on the basis of their phenotypic. Thus, the authors bring novelty in their work. However, there are many limitations min this study. Interestingly, authors have mentioned majority of the limitation in the manuscript itself.

Modifications related to the suggestions of the reviewers are highlighted in red font in the revised version of the manuscript.  

Mode of inheritance is one of my major concern for this study which is key limitation. Since the authors is considering genes observed in 2 patients for their analysis and if such genes are from close/ distant relatives; then there will be bias in the findings.

The reviewer is definitely right in underlining this possible bias. However, an analysis that considers also the relation among patients is regrettably not possible using the present VariCarta dataset, since no details about individual (age, gender, ethnicity, family relationship, etc.) is reported in VariCarta. All variants included in VariCarta have been retrieved by the authors of the VariCarta database extracting them from 69 already published papers, without specifying which individuals belonged to the same families. In accordance with this comment of the reviewer we added this aspect in the Methodology (page 3, lines 97-99) and among the limitations of the study (page 13, lines 427-428).

Additionally, I have following suggestions which will made this paper interesting to autism researchers.

We are extremely thankful to the reviewer for suggesting the following analyses. We eagerly performed these analyses and included the results in the new version of the manuscript. We believe that, thanks to these suggestions, the quality of our results has improved, leading even to identify new biological processes that we had not observed in the previous analyses.

  1. Can you divide the genes into groups on the basis of CADD (<10, 10-20, 20-30, >40) and do the same analysis.?

We performed the suggested analysis and presented the results by adding a completely new paragraph to the manuscript. Please, refer to paragraph 3.5 (page 11, lines 325-338), as well as to page 11, lines 365-367, and to page 13, lines 429-434. The complete list of variants by CADD subgroups has been added to the manuscript as an Additional File (Supporting Information File 3).

  1. Do the same for allele frequency <0.01 vs <0.05.

We performed the suggested analysis and presented the results by adding another completely new paragraph to the manuscript. Please, refer to paragraph 3.4 (page 10, lines 312-323), as well as to page 12, lines 373-378, and to page 13, lines 429-434. The complete list of variants by allele frequency has been added to the manuscript as an Additional File (Supporting Information File 3)

  1. Similarly divide the genes based on the pattern of inheritance de novo vs inherited?

In the specific case of the VariCarta dataset, it was not possible to perform a significant comparison between de novo and inherited variants, as most of the variants retrieved from literature by the authors of VariCarta were de novo variants (99.7%). Nevertheless, in order to estimate the possible impact of inherited variants in computing similarity between patients, we computed a similarity matrix based only on inherited variants (0.3% of all variants). The result showed very small clusters with leak similarity values between patients, while a clustering considering only the de novo variants (99.7% of all variants), yielding the same results we had obtained without excluding the inherited ones. All these considerations have been added to the methodology (page 3, lines 116-125) and results (page 6, lines 247-258).

 If the findings don’t change with these factors; then it will be interesting to the researchers.

When adding these analyses our findings were confirmed, but were also enlarged by adding some biological processes related to learning, which are – of course - of great importance to ASD. Thank you again to the reviewer.

Reviewer 2 Report

In this manuscript from Emberti et al, the authors apply a gene clustering analysis to variants identified via whole-genome sequencing in a large autism spectrum disorder (ASD) cohort with the goal of identifying novel gene networks in autism.

Overall it is unclear how the analysis is controlled and how the variants are classified and whether the presented results are sound.

1) Does the analysis account for population stratification and some of the groups coming from specific datasets? How do we know that these variants do not also exist in control populations? The effect of the chosen variants is uncertain and this reduces the reliability of the follow-up analyses.

2) The authors refer to the variants as mutations, but it is unclear whether they would have any effect on protein function, especially as it pertains to synonymous SNVs.

3) Many of the shared gene categories have appeared in some shape or form in multiple genetic analyses in humans and RNAseq studies of animal models of ASD. It is overall already accepted that genes involved in neuronal function, neuronal differentiation and synaptogenesis are involved in the pathogenesis of ASD and it is unclear how this analysis provides additional information. A better listing and description of the cluster-specific changes in the results may be helpful as the authors only briefly mention them in the Discussion.

Author Response

In this manuscript from Emberti et al, the authors apply a gene clustering analysis to variants identified via whole-genome sequencing in a large autism spectrum disorder (ASD) cohort with the goal of identifying novel gene networks in autism.

 Overall it is unclear how the analysis is controlled and how the variants are classified and whether the presented results are sound.

 Modifications related to the suggestions of the reviewers are highlighted in red font in the revised version of the manuscript.  

1) Does the analysis account for population stratification and some of the groups coming from specific datasets? How do we know that these variants do not also exist in control populations? The effect of the chosen variants is uncertain and this reduces the reliability of the follow-up analyses.

Of course, these points raised by the reviewer are of great importance.

However, an analysis based on population stratification is regrettably not possible using the present VariCarta dataset, since no details (age, gender, ethnicity, family relationship, etc.) about individuals are reported in VariCarta. Only variants retrieved from the individuals are reported. This is definitely a limitation of the analyzed dataset (and not so much of the used methodology). Nevertheless, we tried to plainly highlight this in the methodology (page 3, lines 97-99) and among the limitation of our study (page 13 lines 427-428).

All variants included in VariCarta have been retrieved by the authors of the VariCarta database extracting them from 69 already published papers. All these papers, associated these variants with ASD after having compared individuals with ASD with neurotypical controls. The variants that are included in VariCarta are therefore already the result of controlled studies. The variants belonging to controls have not been included in VariCarta. This is specified by the authors of VariCarta, when – in the presentation of their dataset - they state: “At this stage, we exclude variants that are found in control subjects or subjects not reported to have an ASD diagnosis”. We now better clarified that point in the methodology (page 3, lines 111-114), we analyzed it (page 6, lines 253-258) and discussed it among the limitations (page 13, lines 424-425 and 440-442).  

Actually, the aim of the presented methodology was not so much to identify the effect of one or more variants. Since ASD is likely the combined effect of a set of variants together with external factors and not the result of a single variant, the main aim was to test a methodology, which might help in identifying clusters of individuals characterized by commonalities in their variants. Please, see also below, bullet point 3.

2) The authors refer to the variants as mutations, but it is unclear whether they would have any effect on protein function, especially as it pertains to synonymous SNVs.

Thank you very much for your suggestion about changing the term mutation with the term variant. You are right; the term variant is more appropriate and less confusing. We have now changed the term throughout the whole manuscript.

In any case, in VariCarta the impact of synonymous SNVs variants is limited: considering whole-genome sequencing, there are only 623 synonymous SNVs variants among the 159,279 variants (ie, 0.4%). We also deem that, since the ASD is possibly the result of combined effects of a set of variants together with external factors, even variants that do not have a direct or indirect effect on protein function should be taken into account. Actually, recent important studies claim that even non-coding variants should be taken into account (eg, Zhou et al, Whole-genome deep learning analysis identifies contribution of noncoding mutations to autism risk Nat Genet. 2019 June ; 51(6): 973–980). Stimulated by the comment of the reviewer, we further clarified these aspects in the manuscript, including also the mentioned reference (page 3, lines 132-134; page 12, lines 418-419; page 13, lines 436-438).

3) Many of the shared gene categories have appeared in some shape or form in multiple genetic analyses in humans and RNAseq studies of animal models of ASD. It is overall already accepted that genes involved in neuronal function, neuronal differentiation and synaptogenesis are involved in the pathogenesis of ASD and it is unclear how this analysis provides additional information. A better listing and description of the cluster-specific changes in the results may be helpful as the authors only briefly mention them in the Discussion

The reviewer is right. It is highly plausible that genes involved in neuronal function, neuronal differentiation and synaptogenesis are involved in the pathogenesis of ASD. Also a recent important study (quoted in our manuscript) seems to endorse that (Satterstrom, F.K.; Kosmicki, J.A. et al. Large-scale exome sequencing study implicates both developmental and functional changes in the neurobiology of autism. 2020, Cell, vol. 180, n. 3, pp. 568-584).

However, while we believe that while our study confirms previous findings, it looks at them from a different viewpoint, which – to our knowledge – had not been explored until now.  Since ASD is likely a multifactor disease, where genetic background could be one of the involved factors but not the only one, the main purpose of our study was testing a methodology to identify similarity between ASD individuals, rather than identifying similarities between genes or variants. By applying this methodology, we could identify a set of clusters where the genetic background could impact at different levels. For example, while patients belonging to cluster A present a strong correlation between them and their genetic background seems to have a strong impact on biological processes directly related to ASD, patients belonging to cluster D seem to be genetically quite uncorrelated between them and their variants do not seem to have a significant impact on those processes. Therefore, the idea was to develop a methodology that might allow researchers to better select homogeneous patients’ cohorts for further investigations, so to better appreciate the relative impact of specific biological processes among different autistic phenotypes. We believe that our results might be a useful starting point do develop more focused researches on subgroups of individuals with ASD. We thank the reviewer for the comment, which gave us the possibility to better clarify these aspects in the manuscript. We added to Table 1 and to the Supporting Information File 2 also a brief description of the cluster-specific differences in the main text of the Results (page 8, lines 277-282) and Discussion (page 12, lines 373-378; page 12, lines 393-397; page 12, lines 414-417).

Reviewer 3 Report

I would like to begin by saying that I enjoyed reading your paper.

The writing style is clear, the document is well structured and the methodological approach is sound and well executed.

Nevertheless, I feel there are some issues regarding the initial analysis/data processing steps that might influence the analysis and conclusions.

I feel the main issue with the paper concerns the variant/gene selection step.

As the varicarta database is composed of several variant sets from different populations. there is the possibility that your cluster analysis is indirectly capturing population structure. I would suggest that you address this question and/or acknowledge this limitation. One possiblet way to look into this would be to check the overlap of the different datasets in varicarta with the clusters. I would expect that, in the best case, the datasets will be more or less evenly spread out among the clusters. On the contrary, if the datasets are mainly each in it's own cluster, that might indicate the presence of population structure (or batch effects) in the clusters.

Variant filtering could alleviate this issue, by removing common variants, which contribute more to population structure.

A clarification of this issue would contribute to the robustness of the results.

Also, in line 193 you mention the use of "Chi-square/Fisher’s Exact Test.". Please state the method used.

I have some minor issues with the language:

In line 74, it is not clear to me what is meant by “non-overlapping genes”. Please consider rewriting this sentence.

Throughout the document (for example, in line 133) you use the term "mutative events", which I think is not adequate here. I would suggest you to replace it with "variants" or similar.

Author Response

I would like to begin by saying that I enjoyed reading your paper.

The writing style is clear, the document is well structured and the methodological approach is sound and well executed.

Nevertheless, I feel there are some issues regarding the initial analysis/data processing steps that might influence the analysis and conclusions.

 I feel the main issue with the paper concerns the variant/gene selection step.

Modifications related to the suggestions of the reviewers are highlighted in red font in the revised version of the manuscript.  

As the varicarta database is composed of several variant sets from different populations. there is the possibility that your cluster analysis is indirectly capturing population structure. I would suggest that you address this question and/or acknowledge this limitation. One possiblet way to look into this would be to check the overlap of the different datasets in varicarta with the clusters. I would expect that, in the best case, the datasets will be more or less evenly spread out among the clusters. On the contrary, if the datasets are mainly each in it's own cluster, that might indicate the presence of population structure (or batch effects) in the clusters.

This comment of the reviewer is correct and appropriate. We therefore performed a further analysis to check if the identified clusters could somehow overlap with the different populations from where the VariCarta variants had been retrieved. For each cluster, we identified the papers that contained individuals included in the cluster. We added the result of this analysis at the end of the Additional file 3. In fact, all clusters include populations retrieved from several datasets. There is no cluster, whose original dataset is unique or originating only from few papers. Therefore, we can exclude an overlapping of the original datasets with the identified clusters. However, the reviewer was right in suggesting checking this possibility.

We addressed this issue also in the Results (page 6, lines 253-258)

Variant filtering could alleviate this issue, by removing common variants, which contribute more to population structure.

In VariCarta, an Exome Aggregation Consortium Minor Allele Frequency is unfortunately reported only for a small percentage of variants. Analyzing these variants, only 4% have an ExAC score above 0.05. Therefore, the large majority of these variants are non-common. Furthermore, the authors of VariCarta state that the included variants - retrieved from published studies - had been previously published because they had been considered non-common and therefore had been associated with ASD.

In any case, we now performed also an analysis considering only the variants with allele frequency <0.01 and <0.05. We added a completely new paragraph to the manuscript. Please, refer to paragraph 3.4 (highlighted in red font, page 10, lines 312-323), as well as to page 12, lines 373-378, and to page 13, lines 429-434. The complete list of variants by allele frequency has been added to the manuscript as an Additional File (Supporting Information File 3).

A clarification of this issue would contribute to the robustness of the results.

Also, in line 193 you mention the use of "Chi-square/Fisher’s Exact Test.". Please state the method used.

In fact, it was Fisher’s Exact Test. We corrected it (page 5, line 202).

I have some minor issues with the language:

In line 74, it is not clear to me what is meant by “non-overlapping genes”. Please consider rewriting this sentence.

With non-overlapping genes we meant those genes whose expressible nucleotide sequence does not overlap with the expressible nucleotide sequence of another gene. However, the reviewer is right, the definition might be misleading in this context. As it is not vital for the manuscript, we deleted it from page 2, line 71.  

Throughout the document (for example, in line 133) you use the term "mutative events", which I think is not adequate here. I would suggest you to replace it with "variants" or similar.

Thank you for the suggestion. We replaced “mutative events” with “variants” along the whole text. It is a clearer and unambiguous term.

Round 2

Reviewer 1 Report

Since this study is based on data acquired form public database; there will be certainly some limitation; but the authors have tried to use it with a novel approach and have addressed majority of my previous concerns. I have enjoyed reading the revised version of the manuscript and I don’t have any further queries.

Author Response

We thank again the reviewer for the previous comments, which helped us to improve the quality of the manuscript